# Functional Disability in Adolescents with Chronic Pain: Comparing an Interdisciplinary Exposure Program to Usual Care

**DOI:** 10.3390/children7120288

**Published:** 2020-12-11

**Authors:** Carolien Dekker, Mariëlle Goossens, Bjorn Winkens, Silvia Remerie, Caroline Bastiaenen, Jeanine Verbunt

**Affiliations:** 1Department of Rehabilitation Medicine, Care and Public Health Research Institute, Maastricht University, Universiteitssingel 40, 6229 ER Maastricht, The Netherlands; carolien.dekker@maastrichtuniversity.nl (C.D.); jeanine.verbunt@maastrichtuniversity.nl (J.V.); 2Department of Epidemiology, Care and Public Health Research Institute, Maastricht University, Peter Debyeplein 1, 6229 HA Maastricht, The Netherlands; chg.bastiaenen@maastrichtuniversity.nl; 3Department of Clinical Psychological Sciences, Experimental Psychology, Maastricht University, Universiteitssingel 40, 6229 ER Maastricht, The Netherlands; 4Department of Methodology and Statistics, Care and Public Health Research Institute, Maastricht University, Peter Debyeplein 1, 6229 HA Maastricht, The Netherlands; bjorn.winkens@maastrichtuniversity.nl; 5Rijndam Rehabilitation, Westersingel 300, 3015 LJ Rotterdam, The Netherlands; s.remerie@rijndam.nl; 6Adelante Center of Expertise in Rehabilitation and Audiology, Zandbergsweg 111, 6432 CC Hoensbroek, The Netherlands

**Keywords:** chronic musculoskeletal pain, adolescents, functional disability, multidisciplinary rehabilitation

## Abstract

(1) Background: Chronic musculoskeletal pain (CMP) in adolescents can negatively affect physical, psychological, and social functioning, resulting in functional disability. This randomized controlled trial (RCT) aimed to evaluate the effectiveness of an outpatient rehabilitation program based on graded exposure in vivo (EP) compared with care as usual (CAU: interdisciplinary outpatient rehabilitation care). Both EP and CAU aim to improve functional ability in adolescents with CMP. (2) Methods: Pragmatic multicenter RCT with 12-month follow-up. Adolescents (12–21 years) with CMP were invited to participate. Primary outcome: functional disability; secondary outcomes: perceived harmfulness; pain catastrophizing; pain intensity. Data analysis: intention-to-treat linear mixed model. (3) Results: Sixty adolescents (mean 16 years) were randomized; data for 53 were analyzed. Adolescents in EP showed relevant and significant decreases in functional disability (estimated mean difference at least −8.81, *p* ≤ 0.01) compared with CAU at all times. Significant differences in favor of EP were found for perceived harmfulness at all times (*p* ≤ 0.002), for pain catastrophizing at 2 months (*p* = 0.039) and for pain intensity at 4 and 10 months (*p* ≤ 0.028). (4) Conclusion: EP leads to a significant and clinically relevant decrease in functional disability compared with usual care.

## 1. Introduction

Chronic pain in children and adolescents is a major health concern [1,2]. Chronic musculoskeletal pain (CMP) is one of the most frequently reported pain complaints, next to chronic headache and abdominal pain [1,2]. Internationally, prevalence rates for CMP vary between 4 and 40% and appear to be increasing [1,2]. Since pain during adolescence increases the risk of pain in adulthood [3,4], such complaints need to be treated as early as possible to prevent later suffering.

Evidence about treatment effects of interdisciplinary treatments for adolescent chronic pain is relatively scarce. Psychologically-based treatments, including cognitive behavioral therapy, for adolescent chronic pain appear to be effective in reducing pain, and the quality of studies has improved over the years [5,6]. However, most studies focus on adolescents with headache and use pain reduction as the primary outcome of interest. Evidence of treatment effectiveness on disability and emotional distress in adolescents with CMP is rare [5,6,7].

Pain-related fear can contribute to the development and maintenance of chronic pain. In children and adolescents with chronic pain, Martin and colleagues found that pain-related fear accounted for 39.9% of the variance in pain-related disability [8]. In adults, there is evidence that cognitive behavioral graded exposure in vivo decreases functional disability by reducing pain-related fear [9,10,11]. The fear-avoidance model of chronic pain is the theoretical model underlying exposure therapy [12]. According to this model, in the event of pain, both fear of pain/movement and catastrophic thinking about pain can lead to the development and maintenance of chronic pain problems [12]. By exposing patients to movements and activities previously avoided due to pain-related fear, patients find that normal functioning is possible despite pain [9,10,11].

Recently, more evidence was found on the negative consequences of pain-related fear in children and adolescents with chronic pain [13,14,15]. Meanwhile, the fear-avoidance model has been expanded into an interpersonal fear-avoidance model, incorporating interactions between adolescents and parents, the social context in which adolescent pain problems arise [16]. These have led to the development of an interdisciplinary graded exposure program (EP), specifically for adolescents with CMP.

The primary objective of this study is evaluating the effectiveness of the EP in reducing functional disability, compared with care as usual (CAU), in adolescents of 12–21 years with CMP who report pain-related fear. Secondary objectives are evaluating the effectiveness of the EP in reducing fear of pain, perceived harmfulness, pain catastrophizing, depressive symptoms, pain intensity, and improving health-related quality of life. Explorative health care utilization and school support and school absenteeism of adolescents until 12 months after treatment in both groups are compared.

We hypothesize that an exposure-based program will be more effective in reducing (pain-related) disability (as measured by the Functional Disability Inventory) in adolescents with CMP who report pain-related fear than usual care. We further hypothesize that adolescents in the EP will utilize less health care and school support, have fewer school absences, and lose less schoolwork in the year after the treatment, compared with usual care.

## 2. Materials and Methods

### 2.1. Study Design

The design was a multicenter pragmatic randomized controlled trial (RCT) to evaluate whether EP is superior to CAU in reducing functional disability, the primary outcome (Cinical Trial Registration number is: NCT02181725). The study protocol is published elsewhere [17]. A pragmatic approach was chosen in line with our clinical focus. With the outcomes of the study, we intend to support clinicians in their deciding between different options for care [18]. Ethical approval was granted for this trial (Project identification number NL47323.068.13) from the Medical Ethical Committee of the Maastricht University Medical Centre. The study was conducted in accordance with the Declaration of Helsinki. All participants in the study gave their informed consent before participation. For the adolescents younger than 18, both adolescents and their parents gave informed consent.

### 2.2. Sample and Procedure

Adolescents were recruited by consultants in rehabilitation medicine in four Dutch rehabilitation centers between August 2014 and September 2016. Patients and their parents were recruited after a pre-treatment screening and after eligibility criteria were checked. Adolescents referred to outpatient rehabilitation treatment for CMP, reporting pain-related fear (in the professional opinion of the interdisciplinary treatment team), aged 12–21 years and with adequate Dutch literacy, were eligible for inclusion. The decision-making process in the evaluation of the presence of pain-related fear was supported by the team’s experience and by the outcome of the Fear of Pain Questionnaire. Exclusion criteria were: any suspicion of a medical (orthopedic, rheumatic or neurological) disease that could fully explain the current severity of pain complaints; any suspicion of an underlying psychiatric disease that would hamper rehabilitation treatment; or pregnancy.

Two of the centers were rehabilitation departments of hospitals offering specialized outpatient rehabilitation care. All centers offered EP and CAU. A coordinator was appointed in each center to support the treatment teams with the study procedures during the trial.

### 2.3. Interventions

The EP consisted of active treatment sessions for both adolescents and parents. This program aimed to restore adolescents’ age-appropriate functional abilities by systematically reducing pain-related fear and catastrophic thinking through gradually exposing adolescents to fear-provoking daily activities and movements, such as bending over, jumping down, cycling, and lifting. The EP had 3 key elements: firstly, education about the fear-avoidance model; secondly, identification of avoided activities using a fear hierarchy; and finally, gradual exposure to anxiety-provoking activities [16]. For adolescents, the treatment entailed an intake session with a consultant in rehabilitation medicine, screening and 14 program sessions of 60 min each, during a 7-week period. Program sessions comprised an interdisciplinary intake session, an education session, and twelve graded exposure in vivo sessions. For parents, three meetings were offered in parallel with their adolescents’ program, in a group or individually. Parent sessions were delivered in a group of 3–6 parent-couples to stimulate interaction between participants. If no additional parent-couples were available, the parent module was delivered to an individual couple to reduce waiting time before starting the program.

For adolescents with hypermobility syndrome, physical training [19,20] was added to the EP to prevent hypermobility problems hindering the graded exposure [6]. For these adolescents, the program incorporated 16 physical training sessions of 120 min each, offered prior to the graded exposure in vivo sessions, expanding program duration to 15 weeks. The modules of the program are presented in the addendum. In addition, a detailed description of the EP is provided in the design article and highlights are again presented in Table 1, with an outline of CAU (graded activity (GA)) [17,21].

CAU, predominantly interdisciplinary cognitive behavioral graded activity treatment, has the aim of restoring adolescents’ age-appropriate functional abilities by encouraging desired behaviors, and a time-contingent, stepwise increase in activity levels. For the control intervention, centers followed their own CAU protocol, based on a consensus document of the Dutch working group for youth with chronic pain and fatigue. CAU treatment duration varied between 9–16 weeks, according to each center’s practical and logistical constraints,

Both interventions were of specialized rehabilitation care offered by interdisciplinary treatment teams consisting of a consultant in rehabilitation medicine, a psychologist, and a physiotherapist or occupational therapist. In CAU, a social worker might also be involved. In both arms, adolescents were asked to refrain from other (co-)interventions, and medication use was reduced or terminated if possible. In both EP and CAU, an individual treatment plan was proposed to adolescents and parents, the teams evaluated progress regularly and the consultant in rehabilitation medicine evaluating progress with the adolescents on their treatment. A detailed description of all elements (differences and similarities) included in both treatment programs is presented in the design article for this study, published in 2016 [6].

### 2.4. Measurement Points, Describing Baseline, and Outcome Measures for Treatment Effectiveness

Measurements were at baseline and at 2, 4, 10, and 12 months after start of EP, by digital questionnaires, accessible through a personalized link sent by email. Monthly diaries to assess health care utilization, school support and school absenteeism were used after the end of the treatment for a period of 12 months. Description of the measures and details of their psychometric properties are published in the design article for this study [17].

The primary outcome was functional disability, measured with the Functional Disability Inventory (FDI, 15 items, scored on a 0–4 point Likert scale: range 0–60, higher scores indicating more severe disability) [22,23]. Secondary outcomes were fear of pain (Fear of Pain Questionnaire) [24,25], perceived harmfulness (Photograph Series of Daily Activities for adolescents) [26], pain catastrophizing (Pain Catastrophizing Scale) [27], depressive symptoms (Children’s Depression Inventory) [28], pain intensity (Visual Analogue Scale) [29], and pain-specific quality of life (Quality of Life Questionnaire for Adolescents with Chronic Pain) [30].

### 2.5. Protocol Adherence and Contamination Check

An adapted Method of Assessing Treatment Delivery (MATD) was used [21,31] to measure protocol adherence in EP and verify that neither intervention was contaminated with elements of the other. Protocol adherence was defined as the degree to which essential treatment elements of EP were offered by the treatment teams [31,32]. Treatment teams recorded their own program sessions. A random sample of 36 audio- and video-recorded sessions (14% of 262 recorded sessions) in the four different settings was drawn for analysis by one of the researchers (CD). Outcomes were reported as percentages of protocol adherence and treatment contamination.

### 2.6. Randomization, Allocation Concealment, and Blinding

Minimization was used to reduce imbalance in treatment groups, factors chosen being age, sex and treatment center. In each center, the first adolescent had a 50% probability of being allocated to EP or CAU. In case of an imbalance in minimization factors, the probability of allocation to a particular group was adjusted to 90% for each following adolescent, to better ensure balance. The procedure was executed by a validated electronic randomization system (ALEA, offered by the Clinical Trial Center Maastricht). After written informed consent, the site coordinator inserted participant data, the system then randomizing the adolescent and arranging blinded treatment allocation. The randomization and concealed allocation process included blinding of all relevant caregivers, and of statisticians in the trial. Patients were kept naïve about the preference of the researcher regarding the interventions. Data collection and analysis remained blinded until results were analyzed.

### 2.7. Sample Size

Sample size was calculated for the primary outcome measure (FDI). A mean of 23 points (SD = 9.2) (own unpublished clinical data) and expected mean difference of 5 points (approximately 25% difference) between groups on the total average FDI score at the end of treatment were used. Given α = 0.05, two-sided testing, a power of 80%, and anticipating 15% loss to follow-up, a sample size of 62 participants per trial arm, 124 participants in total, was calculated.

### 2.8. Statistical Analysis

Descriptive statistics were used to explore the data, check for outliers and summarize baseline characteristics (number, % or observed mean, SD) for the adolescents. Analyses were performed in IBM SPSS Statistics for Windows, version 25 (IBM Corp.: Armonk, NY, USA).

### 2.9. Analysis of Treatment Effectiveness

To evaluate effectiveness, intention-to-treat linear mixed model analysis was used. This analysis accounts for correlation between repeated measures, uses all available data, assumes missing values to be random (missing at random, MAR), and corrects for baseline differences. Since it uses a likelihood approach, no imputation strategy was used. The primary and secondary outcome measures were used as dependent variables, while time (categorical: 0, 2, 4, 10, and 12 months), group (intervention vs. control), interaction between time and group, and minimization variables (age, sex, and center) were included as fixed factors. If necessary, variables related to missing outcome values were included in the fixed part of the model to ensure MAR. As for the random part of the model, several options were considered, including an unstructured (UN) covariance structure for repeated measures, or a random intercept and/or random slope model (unstructured or variance components). The model with the smallest Bayesian Information Criterion (BIC) was chosen to be the best fitting model. Effect sizes are reported as estimated mean differences with 95% confidence interval (CI) between intervention and control. As a sensitivity analysis, the linear mixed model analysis was repeated, excluding adolescents with hypermobility syndrome from both EP and CAU, as the amount of active treatment sessions in the EP would be different for this group of adolescents. Two-sided *p*-values ≤ 0.05 were considered statistically significant.

### 2.10. Analysis of Treatment Delivery

The recordings of treatment sessions were scored by two independent raters, a master’s student in developmental psychology and a health scientist. They were trained to analyze the recordings for protocol adherence. Where inter-rater reliability was sufficient (Cohen’s kappa ≥ 0.61) [33], mean scores of both raters were used for subsequent analysis. Following the criteria of Leeuw and colleagues [31], in the EP, for sufficient protocol adherence, the proportion of essential treatment elements present in three different program phases (preparation, education, treatment) should exceed 70%. Contamination was considered absent when less than 10% of prohibited treatment elements were found in both treatments. Furthermore, more than 90% of the recorded sessions should be classified correctly as belonging to either EP or CAU.

## 3. Results

### 3.1. Description of the Study Population

Seventy-seven eligible adolescents were invited to participate. Seventeen participants declined for different reasons. Sixty adolescents were randomized but, because of seven completely missing cases, data from 53 were analyzed (Figure 1). Since the number of participants did not progress as planned, the recruitment period was extended by 7 months from the planned 18 months. After the extended recruitment period, the study had to be terminated for financial and logistical reasons although the intended number of participants had still not been reached.

In Table 2, adolescent characteristics and baseline scores on the outcome measures are reported. Mean age was 16.0 years (SD = 1.87, range 12–20), 49 (92%) adolescents were female. At baseline, there were no meaningful differences between the groups. Ten adolescents were identified as having a hypermobility syndrome: 6 were randomized to EP and received an additional physical training program as part of the new treatment and 4 were randomized to CAU and received usual care.

### 3.2. Effects of the Multimodal Rehabilitation Program

Table 3 shows treatment effects of EP compared with CAU. No variable was significantly related to missing values in the outcome measures at any time point. For all dependent variables, a random intercept model gave the best fit.

For the primary outcome, FDI, estimated mean differences of at least 8.8 points (*p* ≤ 0.011) between EP and CAU, in favor of EP, were observed for all time points, corrected for baseline (Table 3, Figure 2).

For secondary outcomes, significant differences in favor of EP were found for perceived harmfulness at all time points (*p* ≤ 0.002), for pain catastrophizing (PCS) at 2 months follow-up (*p*-value = 0.039), for depressive symptoms at 10 months follow-up (*p* = 0.008), for pain intensity at 4 and 10 months follow-up (*p*-value ≤ 0.028), for quality of life Psychological Functioning domain at 2 and 10 months follow-up (*p*-value ≤ 0.044), and for the Functional Status domain at 2 and 4 months follow-up (*p*-values ≤ 0.016).

An additional analysis was performed excluding adolescents with hypermobility syndrome from both EP (6) and CAU (4). Identical results were found in this analysis.

### 3.3. Health Care Utilization and School Support and Absenteeism

Monthly cost diaries were filled in by 22 adolescents (13 EP and 9 CAU). During the 12 months after treatment, adolescents in EP had fewer hours of contact with general practitioners (EP: M = 0.80; SD = 1.77/CAU: M = 6.58; SD = 9.47), other health care providers (EP: M = 4.28; SD = 7.34/CAU: M = 7.12; SD = 14.72), alternative health care (EP: M = 0.38; SD = 0.96/CAU: M = 1.12; SD = 3.35) and fewer hours of school support (EP: M = 2.03 (SD 0.55); CAU M = 2.83 (SD 4.07)). Adolescents in CAU visited medical specialists less often (EP: M = 2.28; SD = 5.06/CAU: M = 1.66; SD = 2.74). However, adolescents who received EP had more absences from school (M = 45.85; SD = 93.10) compared to those who received CAU (M = 19.31; SD = 39.57). On the other hand, adolescents in CAU missed more hours of self-study and homework (M = 30.25; SD = 60.30) compared to adolescents who followed the EP program (M = 1.74; SD = 6.28).

### 3.4. Protocol Adherence and Contamination

For inter-rater reliability, Cohen’s kappa = 0.69 for the assessment of the treatment elements. Protocol adherence for EP was high since on average 80.8% (SD = 11.05) of the essential treatment elements were present [21]. Contamination was on average 4.9% (SD = 9.19) in EP and 7.7% (SD = 10.30) in CAU, below the threshold and therefore within the acceptable range. Overall, 92% of the recordings were classified correctly as belonging to EP or CAU: one rater misclassified one CAU recording as EP; the other misclassified five CAU recordings as EP.

## 4. Discussions

This study demonstrated that in adolescents with CMP reporting pain-related fear, an interdisciplinary graded EP led to a larger decrease in functional disability than did usual care at all time points. The difference of at least 8.8 FDI points that was found between the groups is statistically significant and clinically relevant [34]. Additionally, the magnitude of this difference was almost twice that predicted during the design of this trial.

Considering the severity of functional disability, adolescents in EP improved on average from moderate to light or no disability. Adolescents in CAU remained, on average, in the moderately disabled category [35]. Furthermore, EP was more effective in decreasing the perceived harmfulness of feared and avoided activities at all time points. At some time points, EP appeared more effective in reducing pain intensity, pain catastrophizing, depressive symptoms, and in enhancing health-related quality of life. Furthermore, adolescents in EP used slightly less health care and school support; however, they were more absent from school, though less from self-study.

The results of this trial add to the evidence on interdisciplinary chronic pain treatment to improve functional ability (e.g., [36,37,38,39,40,41,42]), explicitly focusing on outpatient rehabilitation treatment for adolescents with CMP. To our knowledge, this is the first RCT investigating a graded EP targeting pain-related fear to improve functional ability despite pain. By taking a 12-month follow-up period, the results provide insight into the treatment effects in the longer term, the results showing a slight decrease in the magnitude of the estimated difference between 10 and 12 months. This decrease is difficult to interpret: there were still 22 adolescents completing the questionnaires, while missing questionnaires at these time points could not be related to any measured variable, making selective drop out of the study unlikely. Moreover, the magnitude of the overall decrease still remains well within the clinically relevant change of eight FDI points [34]. Therefore, this decrease in magnitude is not considered to be of significant importance. Furthermore, perceived harmfulness of previously avoided activities and social situations also decreased significantly more in the EP group compared CAU at all time points. For the remainder of the secondary outcome measures, results varied at different time points. No significant differences were found for fear of pain at any time points; pain catastrophizing differed only at 2 months; depressive symptoms showed a difference at 10 months; and pain intensity showed differences at 4 and 10 months. For health-related quality of life, differences in improvement in favor of the EP were visible at 2 and 10 months for the psychological functioning domain and at 2 and 4 months for the functional status domain. No differences were visible for the domains of physical status and social functioning.

In EP, a subgroup of patients, those with hypermobility syndrome, received an additional physical training program. Unfortunately, due to the lower than expected number of participants, no subgroup analysis could be performed, as intended, to study differences in effect size between those who were hypermobile and those who were not. An additional analysis in which hypermobile adolescents were excluded resulted, however, in comparable results, emphasizing the effect of the exposure treatment alone on CMP. Thus, although the hypermobile group received almost double the input in the EP than in CAU, this did not alter the findings.

With a pragmatic approach in this trial, results for the comparative effectiveness of EP and CAU are as close to routine practice as possible [43]. Furthermore, eligibility criteria for referral to outpatient chronic pain rehabilitation for the trial were the same as they were for rehabilitation care outside this study. This increases the external validity of this trial. Internal validity was guaranteed by encouraging treatment teams to adhere to the EP protocol, using randomization with concealed treatment allocation, and blinding of the data collection and analysis [43].

Another strength of this trial was the evaluation of treatment delivery. Although in two of the four centers treatment teams offered either EP or CAU, in the other two centers each team offered both interventions, increasing the risk of contamination by the other intervention. Investigation showed that protocol adherence by the treatment teams in EP was high and contamination was absent, according to the pre-specified criteria. Inter-rater reliability for the rating of the treatment elements was substantial [33]. Protocol adherence was only evaluated for EP. Since CAU was not offered according to the same protocol in all centers, evaluation of protocol adherence was here considered inappropriate.

Some limitations need attention. The specified sample size was not attained. Since the difference between the treatment groups was almost twice as large as the minimum clinically relevant difference used in the sample size calculation, the smaller than desired sample size in this trial was less of an issue. Although this lower inclusion rate did not hinder evaluation of the primary research question, the evaluation of effects related to secondary outcome measures was problematic. Even after a prolonged recruitment period of 25 months, a total of only 60 adolescents were enrolled in the RCT, despite increased efforts to enhance recruitment. These efforts consisted of: prolonging the inclusion period; increasing awareness of the treatment possibilities for adolescents with CMP amongst referring physicians; increasing treatment capacity; raising awareness of treatment possibilities for adolescent CMP in the patients’ association; and publishing information about treatment possibilities in local (medical) monthly magazines. Factors that contributed to the lower recruitment are the fact that adolescents simply declined participation in a scientific study that involved more effort than normal treatment (almost 1/3 of the invited participants declined for various reasons). Further, identification of pain-related fear was found challenging by the newly-trained treatment teams. The most important criterion for offering EP is that pain-related fear be present in the patient [10]. While not explicit in the eligibility criteria, it was implicit in the ‘referral to outpatient rehabilitation’ criterion. It is therefore crucial that pain-related fear be recognized during screening but, for treatment teams inexperienced in screening with a view to EP treatment, this is a challenge. The use of the PHODA-Youth instrument at this stage might offer a solution because this was developed to identify activities or situations perceived as harmful for the painful body part, and which are therefore feared [26]. Furthermore, if pain-related fear is not identified as a (major) problem during the screening, EP is a less appropriate treatment.

Because of the pragmatic approach that was used in the RCT, the results are highly applicable to rehabilitation care outside the study setting and therefore these findings are also relevant for other adolescents with CMP reporting pain-related fear. Due to the diversity in rehabilitation centers participating in the study (two pediatric rehabilitation centers, a rehabilitation department of a general hospital and a rehabilitation department of an academic hospital), the study setting broadly represents actual rehabilitation settings in the Netherlands.

As the consequences of the burden of chronic pain are felt not only by the adolescents themselves, but also by their families and by society as a whole, data from this study may be relevant to them as well. Parents and families are significantly influenced when they care for an adolescent with CMP. Amelioration of adolescents’ complaints benefits parents and families as well. Additionally, as society bears the (large) financial consequences of increased health care utilization due to pain complaints, there is a direct benefit if a treatment results in a reduction of these costs. These costs are of course not only direct and indirect medical costs but include for example productivity losses when parents care for their adolescent sufferers. These costs are of great importance to insurers, policy-makers, and employers. Cost data were only exploratively presented here but should be assessed and evaluated in full in the future.

## 5. Conclusions

In adolescents with CMP, EP leads to a clinically relevant and significantly larger decrease in functional disability than does usual care. Data on protocol adherence and contamination between interventions imply an honest comparison. Therefore, implementation of EP in rehabilitation care for adolescents with CMP and pain-related fear seems a promising way forward. However, further evaluation, such as a full assessment of the cost-effectiveness of the new program, is first recommended.

## Figures and Tables

**Figure 1 children-07-00288-f001:**
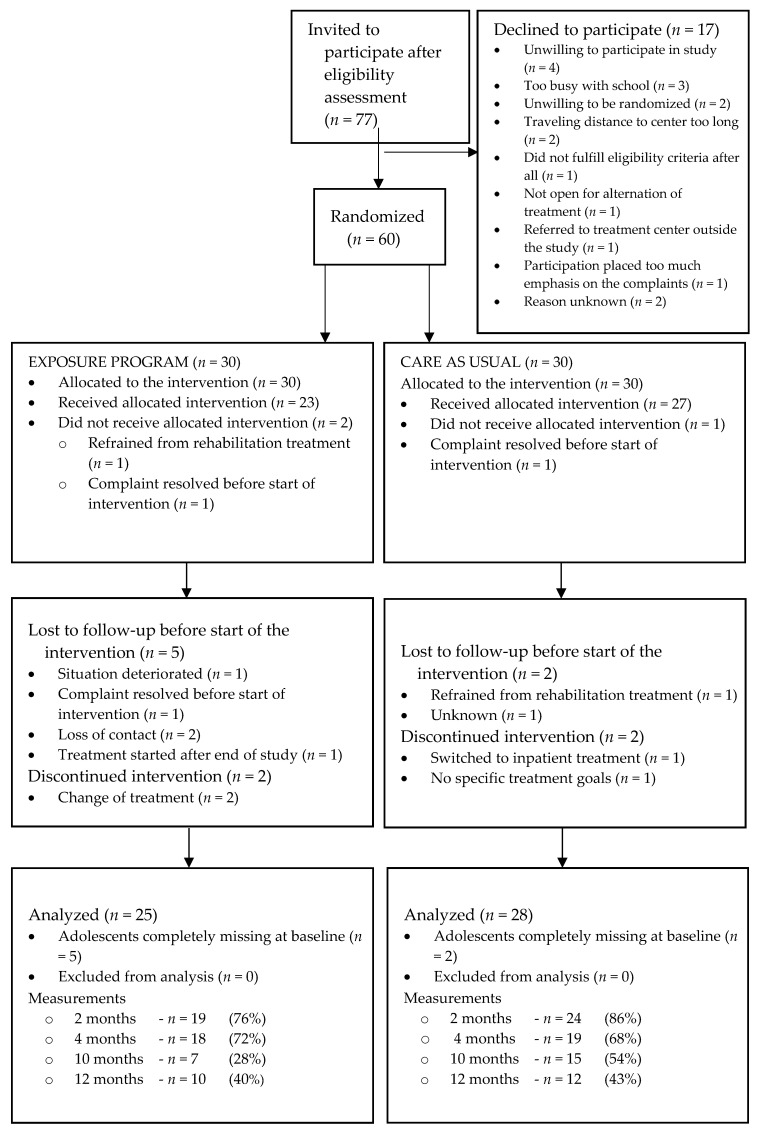
Adolescent flow through the randomized controlled trial (RCT).

**Figure 2 children-07-00288-f002:**
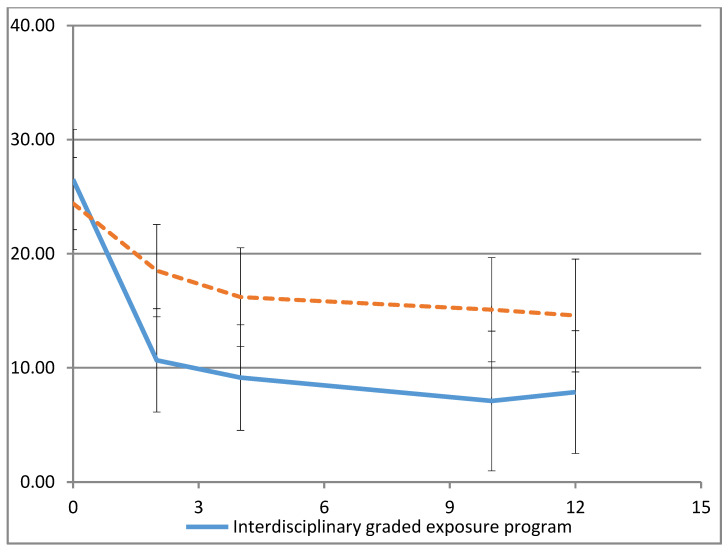
Estimated mean group scores on the FDI at baseline and after 2, 4, 10, and 12 months with 95% CI.

**Table 1 children-07-00288-t001:** Summary of contents of the exposure and care-as-usual programs.

	Exposure Program	Care as Usual (GA)
Underlying paradigm	Classical conditioning; cognitive behavioral	Operant learning principles
Main treatment aim	Restore adolescents’ age-appropriate functional abilities by reducing pain-related fear through gradual exposure to fear-provoking activities	Increase adolescents’ age-appropriate functional abilities by encouraging desired behavior and time-contingent stepwise increase in activity levels
Therapists	Consultant in rehabilitation medicine, psychologist, physiotherapist or occupational therapist	Consultant in rehabilitation medicine, psychologist, physiotherapist or occupational therapist
Number of sessions	1 intake with consultant in rehabilitation medicine + 14 sessions of 1 h. 3 sessions for parents	Variable, from 9 to 16 sessions
Treatment overview in phases	Phase 1: intake + PHODA-youth (1 h): cognitive behavioral analysis of complaints and consequences Phase 2: education (1 h) about treatment rationale, personal fear-avoidance model Phase 3: Exposure with behavioral experiments (12 × 1 h), exposure to fear-provoking activities and movements, generalization and relapse prevention	Phase 1: inventory of the problem Phase 2: problem analyses Phase 3: education Phase 4: choosing activities Phase 5: determining baseline (pain-contingent functioning) Phase 6: determining goal and scheme to increase activity Phase 7: executing scheme, time-contingent increase of activities, encouraging of successful behavior Phase 8: generalization and evaluation
Parent module	3 sessions of 2 h: medical education and treatment rationale, the role of pain in the family system, generalization and relapse prevention	No separate parent program
Additional physical training + alternative treatment schedule	Adolescents with pain complaints related to hypermobility receive 16 (x2 h) physical training focusing on aerobic capacity, muscle strength, core stability, proprioception	No separate program for adolescents with pain complaints related to hypermobility

PHODA-Youth = Photograph Series of Daily Activities—Youth.

**Table 2 children-07-00288-t002:** Characteristics of study participants at baseline (*n* = 53, *n* = 7 missing).

	Exposure Program (*n* = 25)	Care as Usual (*n* = 28)	Total (*n* = 53)
Age (years) − mean (SD)	15.9 (1.99)	16.2 (1.79)	16.0 (1.87)
Sex (female) − *n* (%)	24 (96)	25 (89)	49 (92)
Relative with pain complaints − *n* (% *)	13 (62) (4 missing)	15 (60) (3 missing)	28 (61)
Other health issues − *n* (% *)	8 (38) (4 missing)	11 (44) (3 missing)	19 (41)
Onset of current pain complaints − *n* (% *)	(4 missing)	(4 missing)	(8 missing)
<1 year ago	5 (24)	12 (50)	17 (38)
1–5 years ago	14 (67)	11 (46)	25 (56)
>5 years ago	2 (10)	1 (4)	3 (7)
Problems with sleep − *n* (%)	14 (67) (4 missing)	17 (68) (3 missing)	31 (67) (7 missing)
Education − *n* (% *)	(3 missing)	(3 missing)	(6 missing)
Low	11 (50)	16 (64)	27 (58)
Middle	5 (23)	6 (24)	11 (23)
High	6 (27)	3 (12)	9 (19)
Absence at school in the past year − *n* (% *)	(3 missing)	(3 missing)	(6 missing)
0–14 days	14 (64)	15 (60)	29 (62)
15–30 days	3 (14)	1 (4)	4 (9)
1–3 months	2 (9)	6 (24)	8 (17)
4–6 months	2 (9)	1 (4)	3 (6)
7–12 months	1 (4)	2 (8)	3 (6)
FDI (scored 0–60) − mean (SD)	24.7 (10.3)	23.1 (8.1)	23.8 (9.1)
QLA-CP (scored 0–3) − mean (SD)			
Domain Psychological Functioning	1.57 (0.47)	1.67 (0.51)	1.62 (0.49)
Domain Functional Status	1.74 (0.53)	1.86 (0.44)	1.80 (0.48)
Domain Physical Status	1.81 (0.63)	1.76 (0.64)	1.78 (0.63)
Domain Social Functioning	1.72 (0.60)	1.81 (0.59)	1.77 (0.59)
FOPQ (scored 0–96) − mean (SD)	40.1 (16.7)	38.7 (13.7)	39.3 (15.0)
PCS-C (scored 0–52) − mean (SD)	22.1 (11.0)	20.3 (9.5)	21.1 (10.2)
CDI (scored 0–54) − mean (SD)	26.1 (2.55)	25.7 (2.53)	25.9 (2.51)
VAS (0–100) − mean (SD)	53 (14)	55 (22)	54 (18)
PHODA-Youth (scored 0–510) − mean (SD)	191 (121)	180 (119)	185 (119)
Credibility (CEQ, scored 3–27) − mean (SD)	17.7 (5.1)	18.3 (5.2)	18.0 (5.0)
Expectancy (CEQ, scored 2–18) − mean (SD)	13.2 (2.6)	12.5 (3.5)	12.8 (3.1)

Note. * Valid percent. FDI (Functional Disability Index) = functional disability; QLA-CP (Quality of Life Questionnaire for Adolescents with Chronic Pain) = quality of life; FOPQ (Fear of Pain Questionnaire) = pain-related fear; PCS-C (Pain Catastrophizing Scale-Child version) = pain catastrophizing; CDI (Child Depression Inventory) = depressive symptoms; VAS (Visual Analogue Scale) = pain intensity; PHODA-Youth (Photograph Series of Daily Activities-Youth) = perceived harmfulness.

**Table 3 children-07-00288-t003:** Results of Linear Mixed Model analyses for all outcome measures (*n* = 53 at baseline).

	Estimated Mean Difference * (95% CI); *p*-Value			
	At 2 months (*n* = 43)	At 4 months (*n* = 37)	At 10 months (*n* = 22)	At 12 months (*n* = 22)
FDI	−9.96 (−15.39 to −4.53); 0.000	−9.16 (−14.79 to −3.52); *0.002*	−10.09 (−17.17 to −3.01); *0.006*	−8.81 (−15.59 to −2.044); *0.011*
FOPQ	−3.61 (−12.60 to 5.37); 0.427	−8.00 (−17.26 to 1.26); 0.090	−6.43 (−18.16 to 5.31); 0.280	−8.08 (−19.21 to 3.044); 0.153
PHODA-Youth	−82.19 (−131.06 to −33.32); *0.001*	−108.62 (−159.21 to −58.03); *0.000*	−134.32 (−203.17 to −65.46); *0.000*	−96.12 (−157.63 to −34.61); *0.002*
PCS-C	−5.86 (−11.42 to −0.30); *0.039*	−4.96 (−10.75 to 0.83); 0.092	−4.89 (−12.15 to 2.37); 0.185	−5.58 (−12.46 to 1.31); 0.112
CDI	−1.57 (−5.04 to 1.90); 0.371	−1.14 (−4.72 to 2.44); 0.530	−6.16 (−10.70 to −1.62); *0.008*	−3.27 (−7.57 to 1.03); 0.135
Pain intensity	−11.80 (−24.70 to 1.10); 0.073	−14.88 (−28.08 to −1.67); *0.028*	−21.94 (−39.76 to −4.13); *0.016*	−10.74 (−26.79 to 5.30); 0.187
QLA − Psychological Functioning	5.55 (0.15 to 10.95); *0.044*	3.90 (−1.71 to 9.51); 0.171	7.58 (0.49 to 14.66); *0.036*	4.94 (−1.79 to 11.68); 0.149
QLA − Functional Status	3.96 (1.12 to 6.80); *0.007*	3.63 (0.68 to 6.58); *0.016*	3.71 (−0.01 to 7.43); 0.051	3.52 (−0.01 to 7.06); 0.051
QLA − Physical Status	0.62 (−1.51 to 2.75); 0.567	0.66 (−1.56 to 2.87); 0.559	0.77 (−2.01 to 3.55); 0.585	1.02 (−1.64 to 3.68); 0.448
QLA − Social Functioning	−0.25 (−4.55 to 4.05); 0.909	2.88 (−1.30 to 7.07); 0.176	2.74 (−2.85 to 8.33); 0.334	1.97 (−3.37 to 7.31); 0.467

Note. * corrected for baseline, center, age and sex (Random Intercept model). *p*-value in italics = statistically significant ≤ 0.05. FDI (Functional Disability Index) = functional disability; FOPQ (Fear of Pain Questionnaire) = pain-related fear; PHODA-Youth (Photograph Series of Daily Activities-Youth) = perceived harmfulness; PCS-C (Pain Catastrophizing Scale-Child version) = pain catastrophizing; CDI (Child Depression Inventory) = depressive symptoms; QLA-CP (Quality of Life in Adolescent with Chronic Pain).

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
