# Peer review of "Functional Disability in Adolescents with Chronic Pain: Comparing an Interdisciplinary Exposure Program to Usual Care"

_children, 2020, doi:10.3390/children7120288_

Round 1

Reviewer 1 Report

This is a well-designed study which attempts to further refine the key elements of an outpatient program for the rehabilitation of chronic musculoskeletal pain in youth.  The authors compare the outcomes in children with significant fear of pain using either care as usual (CAU) which is somewhat variable among the 4 institutions in the study to a highly developed exposure program (EP).  Their primary outcome measure is the Functional Disability Index but they looked at a number of other variables as well over a 12-month follow-up period.  They found that the EP group had significantly less functional disability at all measured points throughout the study and differences in pain intensity and pain catastrophizing at specific time periods.  They conclude that for the population of youth with widespread musculoskeletal pain and fear of pain, an exposure approach is better than the more traditional cognitive behavioral approaches. 

In the course of their study, the authors encountered a number of complexities which the acknowledge.  Primarily, they had problems recruiting the desired sample despite extending the length of recruitment by 7 months.  Thus, they only recruited 50% of the desired sample size but due to the strength of their results had enough power to support their hypotheses.   Another issue they address is the issue of hypermobility.  Six of the patients in the EP group and 4 of the patients in the CAU were found to be hypermobile.  However, they received very different programming.  The EP hypermobile group received 14 sessions of physical therapy while the CAU group did not receive any formally.  Ordinarily, this would almost disqualify the findings giving the almost doubling of input into the EP group that was not related to the primary intervention but the authors report, that they removed the hypermobile group from their calculations and thebe a findings were identical.  Although the authors report this in the results, I think it is important to mention in the discussion.

This paper would benefit if a number of issues were addressed:

  • This paper specifically addresses a specific intervention for youth with fear of pain. It would be helpful to the general reader if the authors report what percentage of individuals with chronic pain this represents.
  • Because the interventions were offered at 4 different settings, it would be helpful to know that there were similarly administered at each of the institutions. In particular, the authors should report on the reliability of the EP intervention across sites.
  • The authors do not address the impact of the 3 family sessions that the EP group had on the results. In both groups, approximately 60% of the sample had a relative with chronic pain.  Although not specified, this is likely a parent in a significant percentage of the time.  The impact of the parents on chronic pain is well recognized and parent involvement is typically a cornerstone of any treatment approach.  This difference should clearly addressed in the discussion. 
  • Although the authors discuss in some detail the theory behind exposure therapy and even lay out the agenda for its use, it would be helpful to those not familiar with this approach to hear some specific examples of “fear provoking activities”.
  • I think the discussion should include more discussion of why they think this approach was successful. What is it about exposure theory that is more potent than traditional cognitive behavioral therapy which already has been shown to be quite successful in treating chronic pain.
  • There were a number of grammatical errors and partial sentences throughout the manuscript. To name a few, pg 4, line 130; pg 6 line 205, pg 3 in the discussion, line 390. I recognize that the authors are not primarily English speaking and therefore these errors can be corrected in editing but they should be addressed.

In sum, this is a very well designed and thoughtful paper.  The authors compare 2 groups of youth with chronic musculoskeletal pain and fear of pain using 2 different approaches. One approach is the care as usual which seems quite sophisticated (although it lacks a parent component) which is basically the approach suggested by a Dutch consensus group.  This approach varies somewhat between the 4 institutions involved in the study.  The second approach appears to be a highly manualized approach using EP.  The patients were studied for one year and the authors report significant differences in the groups on their primary outcome, functional disability as well as on some of their secondary outcomes at various time intervals.  This study will provide additional information to those refining the rehabilitative approach to children with chronic pain.

Author Response

We thank the reviewers for their valuable comments to our paper. We also thank the reviewers for the compliments for the study and the article. We hope we have made the requested changes and clarifications to the document. In this letter, we have described the changes. The changes are also highlighted in the article.

In addition, the article was corrected by an English reviewer for English grammar and spelling errors. The corrections made by the English reviewer are not separately marked in the article and this letter.

The following changes and clarifications were made.

Reviewer 1:

  1. We believe that the reviewer may have missed the part about the hypermobile group in the discussion section. However, indeed we did not mention the analysis in the method section. In the discussion we already mentioned the following about the hypermobile group:                              "In EP, a subgroup of patients, those with hypermobility syndrome, received an additional physical training program. Unfortunately, due to the lower than expected number of participants, no subgroup-analysis could be performed, as intended, to study differences in effect size between those who were hypermobile and those who were not. An additional analysis in which hypermobile adolescents were excluded resulted, however, in comparable results, emphasizing the effect of the exposure treatment alone on CMP.”

In the discussion, we added the following to lines 356-357: So, although the hypermobile group received almost double the input of the EP compared to the hypermobile group in the CAU this did not disqualify the findings.

In methods section 2.9 we added the following to lines 205-208: A sensitivity analysis, the linear mixed model analysis was repeated excluding adolescents with hypermobility syndrome from both EP and CAU, as the amount of active treatment sessions in the EP group would be different for this group of adolescents.

2.  We added a percentage of children and adolescents with pain related fear. The following was added to lines 52-54:

(reference: 8. Martin AL, McGrath PA, Brown SC, Katz J. Anxiety sensitivity, fear of pain and pain-related disability in children and adolescents with chronic pain. Pain Research and Management. 2007; 12:267–272.)

In children and adolescents with chronic pain, Martin and colleagues found that pain related fear accounted for of the variance in pain related disability [8].

3. Comment: “Because the interventions were offered at 4 different settings, it would be helpful to know that there were similarly administrated at each of the institutions. In particular the authors should report on the reliability of the EP intervention across sites.

Answer: Indeed the interventions were offered at 4 different settings. A process evaluation was done to check whether the treatments were provided as intended. At section 2.5 ‘Protocol adherence and contamination check’ we already mentioned that we measured protocol adherence of the EP. This means that we verified whether the essential treatment elements were indeed offered. At the results, section 3.4, the results of the protocol adherence was described.

We however did not mention that the process evaluation was done in the different teams at 4 different locations.

We made the following addition to section 2.5 (lines 169-170):

A random sample of 36 audio- and video-recorded sessions (14% of 262 recorded sessions) in the four different settings was drawn for analysis by one of the researchers (CD).

4. The reviewer asked for specific examples of fear provoking activities. We now added the following (lines 108-109):  

.. such as bending over, jumping of something, cycling and lifting something.

5. The reviewer asked ‘What is it about exposure theory that is more potent than traditional cognitive behavioral therapy which already has been shown to be quite successful in treating chronic pain.”

At lines 109-111 we added the following: The EP program had 3 key elements; firstly education regarding the fear avoidance model, secondly identification of avoiding activities using a fear hierarchy and finally gradual exposure to anxiety provoking activities [16].  

Reviewer 2 Report

The submitted manuscript will likely be of interest to providers working with youth with ongoing pain. Overall, the study design and results are clear and valuable; however, there are a few points that should be addressed:

1) Throughout the manuscript there are grammatical errors, including changes in tense and verb use, as well as unnecessary words within sentences. I recommend that the authors have a native English speaker/reader review the manuscript to provide all relevant feedback. there are also words within the manuscript which have not been translated into English, for example: 'van' in Table 1

2) Line 49 - the comment about pain in adolescence increasing the risk of pain in adulthood seems unfinished. While further information is provided later in the introduction, the comment at the end of the first paragraph does not seem sufficient for the point. I would recommend removing it from that paragraph, or adding more relevant content in the first paragraph about why that matters

3) line 118 - it seems that there is a word missing between "to the" and "to prevent" - see comment above as well

4) lines 266- 269 should be moved to be at the end of line 219.

5) line 310 - since there is contamination, it cannot be absent, but it can be in the acceptable range

6) A major concern regarding the intervention is that it was not acceptable to 1/3 of the youth population; therefore, the possible clinical relevance of the intervention is in question. This should be addressed more thoroughly in the discussion. Specifically, how is this higher intensity intervention useful if youth are not interested in participating

7) lines 390-394 - it is unclear what the purpose of these sentences are in this section of the manuscript.

8) the conclusion does not provide any clinical relevant information from the results. It would be more useful to include additional information about changes to outcome measures than to the protocol adherence and cost-effectiveness.

Author Response

Reviewer 2

  1. We thank the reviewer for the comment on grammatical errors. As recommended a native English reader reviewed our manuscript.
  2. The reviewer recommended adding more relevant information to the first paragraph. The following information was added at lines 44-45:

    ‘the complaints need to be treated as early as possible, to prevent long term suffering’.

  3. There was indeed a word missing in line 119. The sentence was corrected: For adolescents with hypermobility syndrome, physical training [18, 19] was added to the EP to prevent hypermobility problems hindering the graded exposure [6].
  4. Lines 266-269 were removed and placed at the end of line 225-228 (at the end of the paragraph):  “Since the number of participants did not progress as planned, the recruitment period was extended by 7 months from the planned 1.5 years. After the extended recruitment period of 7 months, the study had to be terminated for financial/logistic reasons although the intended number of participants had still not been reached. “
  5. We agree with the reviewer that (Line 310) since there is contamination, it cannot be absent, but it can be in the acceptable range. We reformulated the sentence at line 316: Contamination was on average 4.9% (SD=9.19) in EP and 7.7% (SD=10.30) in CAU, below the threshold and therefore within the acceptable range.
  6. With this comment the reviewer suggests that the youth is not interested in participating.We do not agree with the reviewer. It is important to consider, that the reasons for non-participation in this stud did not reflect non-willingness for participation in the treatment. As a main reason for non-participation the expected burden of participating in a scientific study (including completing questionnaires at several time-points) was mentioned. During adolescents, willingness to participate in a scientific study can be less as compared to other phases in life. (Tsevat et al., 2020).

    Tsevat RK, Radecki Breitkopf C, Landers SE, de Roche AM, Mauro C, Ipp LS, Catallozzi M, Rosenthal SL. Adolescents' and Parents' Attitudes Toward Adolescent Clinical Trial Participation: Changes Over One Year. J Empir Res Hum Res Ethics. 2018 Oct;13(4):383-390.

  7. The purpose of the lines 390-394 was unclear to the reviewer. The reviewer is indeed right. We already mentioned the reference earlier in the discussion. We removed these lines.
  8. The reviewer asked for more information about changes in outcome measures in the conclusion. However, we decided to mention only the information on functional disability, since this is the primary outcome measure as formulated in the research question. We suppose the conclusion would be too extensive if more outcome measures were mentioned.
